# Iso-Mukaadial Acetate from *Warburgia salutaris* Enhances Glucose Uptake in the L6 Rat Myoblast Cell Line

**DOI:** 10.3390/biom9100520

**Published:** 2019-09-22

**Authors:** Nontokozo Z. Msomi, Francis O. Shode, Ofentse J. Pooe, Sithandiwe Mazibuko-Mbeje, Mthokozisi B. C. Simelane

**Affiliations:** 1School of Life Sciences, University of KwaZulu-Natal, Durban, Westville 4000, South Africa; ntokozomsomi0@gmail.com (N.Z.M.); PooeO@ukzn.ac.za (O.J.P.); 2Department of Biotechnology and Food Technology, Durban University of Technology, P.O. Box 1334, Durban 4000, South Africa; francisshode@gmail.com; 3Biomedical Research and Innovation Platform, South African Medical Research Council, Tygerberg 7505, South Africa; Sithandiwe.Mazibuko@mrc.ac.za; 4Division of Medical Physiology, Faculty of Health Sciences, Stellenbosch University, Tygerberg 7505, South Africa; 5Department of Biochemistry, Faculty of Science, University of Johannesburg, Auckland Park 2092, Johannesburg 2006, South Africa

**Keywords:** *Warburgia salutaris*, iso-mukaadial acetate, glucose utilisation, antidiabetic activity, AMPK, AKT

## Abstract

Diabetes mellitus (DM) is a chronic metabolic disorder which has become a major risk to the health of humankind, as its global prevalence is increasing rapidly. Currently available treatment options in modern medicine have several adverse effects. Thus, there is an urgent need to develop alternative cost-effective, safe, and active treatments for diabetes. In this regard, medicinal plants provide the best option for new therapeutic remedies desired to be effective and safe. Recently, we focused our attention on drimane sesquiterpenes as potential sources of antimalarial and antidiabetic agents. In this study, iso-mukaadial acetate (Iso) (**1**), a drimane-type sesquiterpenoid from the ground stem bark of *Warburgia salutaris*, was investigated for glucose uptake enhancement in the L6 rat myoblast cell line. In vitro assays with L6 skeletal muscle cells were used to test for cytotoxicity, glucose utilisation, and western blot analysis. Additionally, the inhibition of carbohydrate digestive enzymes and 1,1-diphenyl-2- picrylhydrazyl (DPPH) scavenging activity were analysed in vitro. The cell viability effect of iso-mukaadial acetate was the highest at 3 µg/mL with a percentage of 98.4. Iso-mukaadial acetate also significantly and dose-dependently increased glucose utilisation up to 215.18% (12.5 µg/mL). The increase in glucose utilisation was accompanied by enhanced 5’ adenosine monophosphate-activated protein kinase (AMPK)and protein kinase B (AKT) in dose-dependent manner. Furthermore, iso-mukaadial acetate dose-dependently inhibited the enzymes α-amylase and α-glucosidase. Scavenging activity against DPPH was displayed by iso-mukaadial acetate in a concentration-dependent manner. The findings indicate the apparent therapeutic efficacy of iso-mukaadial acetate isolated from *W. salutaris* as a potential new antidiabetic agent.

## 1. Introduction

Diabetes mellitus is a metabolic disorder characterized by inappropriate hyperglycaemia, which is stimulated by several factors such as inadequate insulin secretion, insulin inaction, and at times, both [1,2]. This disorder prevails worldwide, with its occurrence increasing at an alarming rate [3]. The treatment options for DM are based on parental insulin and oral antidiabetic drugs. These hypoglycaemic agents include biguanides, sulphonylureas, and other drugs like acarbose [4]. However, side effects that have been reported from the use of these drugs include hepatorenal disturbances, hypoglycaemic coma [5], acute hepatitis, diarrhoea, and abdominal pain [6]. The effectiveness of these drugs may also be lost after prolonged usage [7]. As a result, herbal medicines containing a high therapeutic efficacy with minimal adverse effects are favoured [8]. Medicinal plants as antidiabetic agents are very promising [5]. Therefore, there has been a rise in medicinal plant research, studying potential antidiabetic action [5]. 

In the maintenance of postprandial glucose homeostasis, the skeletal muscle plays a role by enhancing glucose uptake from insulin-sensitive muscle cells through the AMPK/p38 MAPK signalling pathway [9]. Postprandial hyperglycaemia culminates in type 2 diabetes and ensues in the formation of advanced glycation end products [10,11]. As such, an important therapeutic approach for diabetes would relate to the achievement of a decrease in postprandial hyperglycaemia [11]. Anti-hyperglycaemic effects of medicinal plants are attributed to their ability to mimic insulin, acting on insulin-secreting beta cells or altering glucose utilization [12]. Medicinal plants which modify glucose utilization act by delaying gastric emptying or delaying glucose absorption [13]. Absorption of glucose can be delayed by reducing the rate of digestion of starch. Inhibition of α-amylase and α-glucosidase, which play a functional role in the process of starch degradation, would delay the absorption process, leading to a decrease in glucose absorption and subsequently reducing postprandial blood glucose levels [14]. Therefore, screening of inhibitors of these enzymes from medicinal plants has gained increase attention in recent years.

Drimane sesquiterpenoids have been reported [15] to have useful biological activities. The main aim of this study was to investigate the antidiabetic potential of the crude DCM extract from the stem bark of *Warburgia salutaris* (Bertol.f.) Chiov. (Canellaceae) as well as pure iso-mukaadial acetate, which has been investigated for antimalarial activity [16]. *W. salutaris* is a flowering medium-sized medicinal plant identified as “pepper bark” due to its peppery taste. It is distributed in the Southern Africa region, where it is traditionally used to treat diseases such as diabetes, fever, colds, flu, coughs, and chest infections [17,18]. The phytochemical screening of *W. salutaris* has led to the identification of drimane sesquiterpenes which include: iso-mukaadial acetate [16], salutarisolide [19], muzigadial [20], warburganal, mukaadial, isopolygodial, polygodial, and iso-mukaadial acetate [21]. These phytochemicals possess a wide range of biological activities including antimicrobial, antibacterial, antifungal, piscicidial, antimalarial, and antioxidant properties [16,17]. Drimane sesquitepenoids have also been reported to possess antidiabetic activity [22]. The presence of drimane sesquiterpenes in *W. salutaris* stimulated our research groups interest in isolating compounds from this metabolite class from the plant. In our continued investigation of potential anti-diabetic agents from plants, an evaluation of the compound iso-mukaadial acetate was conducted for potential anti-diabetic activity for the first time and to build rich reservoir of pharmacologically established antidiabetic phytoconstituents with specific references to the novel compounds that are affordable which might be of relevance to other low-income and middle-income countries of the world. Furthermore, our research group performed a scientific validation of the use of *W. salutaris* in traditional medicine.

## 2. Materials and Methods

### 2.1. Extraction and Isolation of Iso-Mukaadial Acetate from W. salutaris

The extraction procedure described in Nyaba et al. [16] was followed. Briefly, the powdered bark of *W. salutaris* (1 kg) was extracted with dichloromethane (DCM) (1:5 *w*/*v*) for three days successively. Whatman (No.1) filter paper was used to filter the crude extract, followed by sample concentration under a reduced pressure (45 °C) to a minimum volume using a Heidolph rotary evaporator (Heidolph Instruments GmbH & CO. KG, Schwabach, Germany). This crude extract was allowed to dry in the fume hood. A portion of the DCM extract was subjected to silica gel column chromatography (60 × 1000 mm; Merck silica gel, 60:0.063–0.200 mm). The eluent system used was hexane: ethyl acetate (8:2), which yielded 35 fractions. The fractions were monitored by Thin Layer Chromatography (silica gel 60 aluminium sheets, F254—Merck, Whitehouse Station, NJ, USA), profiling the fractions into seven combined fractions. The TLC spots were fixed with a solution made of 20% H_2_SO_4_ in methanol, heated for colour development, and visualized under UV light (254 nm). The fractions were left overnight to until desiccation was achieved. A white powder (NN-01) was obtained from the combined fractions 2–4.

### 2.2. Structural Confirmation

The structure of the isolated iso-mukaadial acetate (**1**) was confirmed using nuclear magnetic resonance (NMR) techniques (1H, 13C, HSQC, HMBC, and COSY) and its spectral data were in agreement with values in the literature [16].

### 2.3. Cell Culture and Treatment Conditions

L6 (Cat. No. CRL-10741) were obtained from the American Type Culture Collection (Manassas, VA, USA). These cells were derived from rat skeletal muscle and were sub-cultured and differentiated using a modified method of Venter et al. [23]. Briefly, L6 muscle cells were seeded into 96-well plates (5000 cells/well) for cytotoxicity evaluation and glucose uptake, and 6-well plates (75,000 cells/well) for protein analysis. The L6 cells were maintained in dulbecco’s modified eagle medium (DMEM) supplemented with 10% fetal bovine serum (FCS) at 37 °C in 5% CO_2_ and humidified air for 3 days to achieve 80–90% confluency. Thereafter, the 10% FCS was substituted with 2% horse serum for a further 2 days of culture to facilitate myocytic differentiation, upon myotubule formation, cells were treated with various concentrations of extract and relevant assay were performed.

### 2.4. Preparation of Extracts for Cell Culture

The extracts were freshly made up in cell culture sterile water prior to each assay at a stock concentration of 0.1 mg/µL. Both extracts (iso-mukaadial acetate and crude extract) were then diluted to a concentration of 3, 6, and 12.5 µg/mL in DMEM containing 0.1% bovine serum albumin (BSA); 50 μL of 8 mM glucose containing 1000 ng insulin was used as a positive control.

### 2.5. Cytotoxicity Evaluation

Upon myotubule formation, cytotoxicity assay was determined using crystal violet. Briefly, cells were cultured with fresh medium containing the iso-mukaadial acetate and crude extract compounds at various concentrations (3, 6, 12.5, and 50 μg/mL) for 48 h. Thereafter, culture medium was removed, and cells were fixed with 100 µL of 10% formaldehyde. After removal of the fix solution, 100 μL crystal violet was added and incubated at room temperature for 10 min. The crystal violet dye was removed, and the wells washed three times with tap water and once with distilled water. Plates were left to dry overnight at 37 °C and the bound dye solubilised by adding 200 μL 10% acetic acid. The absorbance at 595 nm was read and the cytotoxicity expressed as a percentage of the untreated control. 

### 2.6. Glucose Utilisation Screening

The glucose utilization in L6 myoblasts cells was determined according to the method described by van de Venter et al. [23]. Cells were seeded and treated identically as described for the cytotoxicity assay, except that after the 48-h treatment and removal of spent culture medium, the cells were washed once with PBS. Then, 50 μL of 8 mM glucose solution (DMEM medium diluted with PBS and supplemented with BSA to a final concentration of 0.1%) were added. For the positive control, 1000 ng of insulin were added. Plates were returned to the incubator. After 2 h, 5 μL were transferred to a new plate and 200 μL of glucose assay reagent (glucose oxidase/peroxidase colorimetric reagent) were added and incubated at 37 °C for 10 min. Absorbance was measured at 510 nm. Glucose utilisation was calculated as the difference between the no cell control and the test sample and expressed as a percentage of the untreated control.

### 2.7. Westen Blot Analyses

Western blot analysis as performed as previously described by Mazibuko et al. [24] with slight modifications. Briefly, after cell culture cells were lysed with cell lysis buffer (RIPA buffer) and 40 µg of protein were loaded in 10% SDS-PAGE gels to separate denatured proteins before being transferred to PVDF-P or nitrocellulose membranes. Fat-fat milk powder in Tris-buffered saline (*w*/*v*) containing Tween 20 was used to block non-specific protein labelling. Thereafter, membranes were labelled overnight at 4 °C with relevant primary antibodies (AKT, p-AKT (Ser473) and AMPK, p-AMPK (Thr172). After overnight incubation, horseradish peroxidase-conjugated secondary antibody was applied for 1.5 h. β-actin was used as the reference control. Thereafter, proteins were detected and quantified by chemiluminescence using a Chemidoc-XRS imager and Quantity One 1-D software (4.6.8, PC), while molecular weight band detection was confirmed using ImageJ software (Bio-Rad Laboratories, 6.0.1), respectively. 

### 2.8. α-Amylase Inhibition Assay

The inhibition of α-amylase was determined according to Sathiavelu et al. [25]. A total of 500 µL of iso-mukaadial acetate and DCM crude extract at varying concentrations and 500 μL of 0.02 M sodium phosphate buffer (pH 6.9 comprising 0.006 M sodium chloride) containing α-amylase solution (0.5 mg/mL) were incubated for 10 min at 25 °C. Afterwards, 500 μL of 1% starch solution in the same buffer were added to each test tube. The reaction mixture was incubated for a further 10 min at 25 °C. 3,5-Dinitrosalicylic acid colour reagent (1 mL) was added to halt the reaction. The test tubes were incubated in boiling water bath for 5 min and cooled to room temperature. Then, 10 mL dH_2_O was added to the mixture, followed by absorbance measurement at 540 nm. Acarbose was used as a positive control. 

### 2.9. α-Glucosidase Inhibition Assay

The inhibition of α-glucosidase activity was done according to the assay adapted from Bajpai et al. [26]. Briefly, 10 μL of test samples at several concentrations and 50 μL of yeast α-glucosidase, dissolved in 100 mM phosphate buffer (pH 7.0) (containing 0.2 g/L NaN_3_ and 2 g/L bovine serum albumin) were mixed in a 96-well micro-plate, and absorbance at 595 nm was read at zero time with a micro-plate reader. After an incubation period of 5 min, 50 μL of *p*-nitrophenyl-α-d-glucopyranoside (5 mM) in the same buffer (pH 7.0) were used as a substrate solution and incubated for a further 5 min at room temperature. Then, 80 μL of 0.2 M sodium carbonate solution were added to terminate the reaction. The absorbance was read at 595 nm. The reaction without α-glucosidase was used as a blank, and acarbose at several concentrations was used as a positive control. 

Each experiment was conducted in triplicates and the enzymatic inhibition rate was calculated as follows:
(1)$$\mathrm{Inhibition}\;\left(\%\right)\;=\frac{\left(\mathrm{Control}\;\mathrm{absorption}-\mathrm{Sample}\;\mathrm{absorption}\right)}{\mathrm{Control}\;\mathrm{absorption}}\;\times 100$$

### 2.10. Antioxidant Activity

The 1,1-diphenyl-2-picrylhydrazyl (DPPH) free radical scavenging effect of iso-mukaadial acetate was determined by a standard method by Brand-Williams et al. [27]. An aliquot of DPPH solution (2 mg in 100 mL MeOH) was prepared and 2 mL of this solution were added to 2 mL of each sample solution in methanol at varying concentrations. The mixtures were left for 30 min in the dark at room temperature. A spectrophotometer was used to measure the absorbance of each sample at 517 nm. The scavenging effect expressed (%) was calculated using the following formula:Scavenging effect (%) = 1 − [A_Sample/_A_contro_] × 100(2)

### 2.11. Statistical Analysis

The data were expressed as means ± SEM. Statistical analysis of cytotoxicity and glucose utilisation was measured using Microsoft Excel software (2013). Statistical comparison of the differences between the means of control and experimental groups was performed with GraphPad Prism Software version 5.00, using one-way analysis of variance (ANOVA).

## 3. Results

### 3.1. Structural Elucidation

The chemical shift of the structure (NN-01) was studied, with the compound regarded as iso-mukaadial acetate (**1**) (Figure 1). The FT-IR spectrum of iso-mukaadial acetate contained characteristic functional group frequencies: IR ν max (3450, 2942, 2848, 1743, 1717, 1691 cm^−1^). The spectroscopic data found similarities to those of cinnamodial (**2**) [18]. However, its melting point of iso-mukaadial acetate (95–98 °C) differed from that of the melting point of cinnamodial (**2**) (135–137 °C) [18]. This difference led to the X-ray crystallographic study of NN-01, which confirmed the structure as iso-mukaadial acetate (**1**) (see Figure 1) [16]. 

### 3.2. Effect of Dichloromethane Crude Extract and Iso-Mukaadial Acetate on Cytotoxic Evaluation and Glucose Utilisation 

The cytotoxicity and glucose utilisation of the DCM crude extract and iso-mukaadial acetate were evaluated in L6 cells. The cell viability of iso-mukaadial acetate was found to be 3 µg/mL with a percentage of 98.4 (Figure 2). Glucose utilisation studies revealed that iso-mukaadial acetate led to a statistically significant glucose uptake (*p* < 0.05), by 129.41% (3.125 µg/mL), 155.22% (6.25 µg/mL), and 215.18% (12.5 µg/mL). However, no apparent enhanced glucose utilisation was observed for the crude extract. Results were compared to insulin, an anabolic hormone which presented a percentage of 171.15% (Figure 3).

### 3.3. DCM Crude Extract and Iso-Mukaadial Acetate Activate the AMPK and AKT Pathway

Both the PI3K/AKT pathway and AMPK play a major role in activating the insulin signalling pathway. Our results showed that iso-mukaadial acetate activated AKT when cells were cultured in low dose (3 µg/mL) no effect was observed under high concentration of (6 and 12 µg/mL). However, with crude extract AKT was only upregulated at a lowest concentration (3 µg/mL) (*p* < 0.001) (Figure 4). In terms of AMPK, both iso-mukaadial acetate and crude extract enhanced AMPK activation in a dose-dependent albeit not significant manner (Figure 5). In both iso-mukaadial and crude extract (12 µg/mL), AMPK activation was more pronounced. 

### 3.4. DCM Crude Extract and Iso-Mukaadial Acetate Inhibit α-Amylase 

Figure 6 shows the variable inhibitory effect of the DCM crude extract and iso-mukaadial acetate on α-amylase activity in vitro. The inhibitory effect of iso-mukaadial acetate was found to be prominent over the crude extract. At the highest concentration of 50 µg/mL, iso-mukaadial acetate showed an inhibitory effect on α-amylase by 41.01%. However, the activity was lower compared to positive standard acarbose.

### 3.5. DCM Crude Extract and Iso-Mukaadial Acetate Inhibit α-Glucosidase 

α-Glucosidase activity was assessed by the release of *p*-nitrophenol from *p*-nitrophenyl-α-d-glucopyranoside in vitro. The activity of the DCM crude extract and iso-mukaadial acetate were presented in Figure 7. The tested samples exhibited various levels of effectiveness in inhibiting α-glucosidase. Iso-mukaadial acetate at the highest concentration 50 µg/mL presented an inhibitory effect of 29.61%. It was observed that positive standard acarbose is a more potent inhibitor of α-glucosidase compared to iso-mukaadial acetate. 

### 3.6. DCM Crude Extract and Iso-Mukaadial Acetate Inhibit DPPH Radical

The DPPH scavenging ability of DCM crude extract and iso-mukaadial was evaluated, with the results are shown in Figure 8. The results showed that the crude extract and iso-mukaadial acetate reduced DPPH radicals in a dose-dependent manner. At a concentration of 50 µg/mL, iso-mukaadial acetate scavenged 42.65% of DPPH radicals. 

## 4. Discussion 

Plants have always been a good source of phytochemicals with therapeutic potential, and currently represent an important pool in ethnopharmacology for the discovery of novel drugs [28]. A pure compound of iso-mukaadial acetate was isolated from the plant *W. salutaris* and evaluated for the first time with respect to its bioactivity as an antidiabetic agent. The use of medicinal plants is presently directed at lowering and controlling blood glucose levels in treating diabetes [29]. Thus, medicinal plants have been proven to enhance glucose uptake by GLUT4 translocation using in vitro glucose model [30]. In the present study, an L6 cell line originally derived from rat skeletal muscle was used. This immortalized myoblast cell line is popular as a model for glucose uptake since the cells differentiate with high reliability into a myotube muscle cell phenotype that expresses the GLUT4 glucose transporter protein naturally [31]. 

A study conducted by Kawabata et al. [32] previously established that triterpenoids isolated from *Ziziphus jujuba* effectively enhanced glucose utilisation through the translocation of GLUT4 transporter. A defect in GLUT4 expression and translocation has been suggested to be the main metabolic irregularity in diabetic skeletal muscle [33]. Furthermore, at higher doses both iso-mukaadial acetate and the crude extract attenuated glucose utilisation. Reduction of cell viability was seen at higher iso-mukaadial acetate concentrations, which may contribute to this observation. Iso-mukaadial acetate was observed to be non-toxic at 3 μg/mL in L6 cells. In the cytotoxicity evaluation, the highest concentration of a test agent should be less than 1000 μg/mL to be considered non-toxic [34]. These findings indicate that antidiabetic activity of iso-mukaadial acetate may be mediated, at least partially through GLUT4 translocation at low concentrations without a toxic insult.

Previous studies have highlighted that the activation of p-AKT is one of the major downstream kinases activated by insulin and is considered to be a vital step in the translocation of GLUT4 during insulin stimulation. In this study, our data indicates that at low doses the crude DCM and iso-mukaadial acetate doses were able to enhance phosphorylation of AKT, thereby increasing insulin sensitivity (Figure 4). Results indicate that the DCM crude and iso-mukaadial acetate increase glucose uptake by activating the PI3K pathway, which is insulin-dependent. Our data further shows that the DCM crude and iso-mukaadial acetate are potent AMPK pathway stimulators in a dose-dependent manner (Figure 5). A study conducted by Tao and colleagues (2010) demonstrated that the expression of AMPK regulates AKT phosphorylation dependent on PI3K and independent of the insulin receptor 1 pathway. Similarly, in this study at high crude DCM and iso-mukaadial acetate doses, the over expression of AMPK may have been important in regulating AKT stimulation (Figure 4 and Figure 5). Furthermore, several studies that have been done using natural compound extracts reported the potential activation of both AKT and AMPK pathways [35,36]. The mechanism whereby AMPK regulates PI3K as well as the underlying molecular mechanism for the antidiabetic observations obtained in this study are yet to be fully conceptualized. In terms of molecular mechanism, our results showed that these extracts activate both insulin-dependent and insulin-independent pathways.

One of the therapeutic approaches in treating DM is to decrease postprandial hyperglycaemia by suppressing the gastrointestinal tract production or absorption of glucose by inhibition of either α-amylase or α-glucosidase enzymes [11]. Therefore, iso-mukaadial acetate and the crude extract were investigated for their inhibitory effect against these key carbohydrate hydrolysing enzymes. Based on the results, iso-mukaadial acetate and DCM crude extract seemed to exert an inhibitory effect on both enzymes. It has been shown in literature that plants have the ability to inhibit the enzymes α-amylase and α-glucosidase, in vitro [11,23,37,38]. Based on these findings, *W. salutaris* can be considered to possess hypoglycaemic effect. 

The malicious effects of diabetes have been also established to be mediated through oxidative stress which arises from the formation of free radicals. Plant derived antioxidant compounds can reduce the generation of free radicals, and thus alleviate diseases triggered by oxidative stress [39]. Iso-mukaadial acetate and DCM crude extract showed the ability to scavenge DPPH radicals in a concentration dependant manner. Similarly, a study conducted by Ai-Dabbas and collaborators [40], showed that the ethanolic extracts and compounds isolated from *V. iphionoides* presented DPPH radical-scavenging activity. Furthermore, a study by Frum et al. [19] reported that the methanol and aqueous extracts of *W. salutaris* exhibited promising antioxidant activities. Therefore, these observations confirm that *W. salutaris* possesses antioxidant activity and may provide protection against oxidative damage in diabetic patients by neutralizing free radicals.

## 5. Conclusions

The present study was aimed to showcase the potential of the plant *W. salutaris* as an anti-diabetic agent. The pure compound, iso-mukaadial acetate, revealed low cytotoxicity with a significant effect on glucose utilisation in vitro. Thus, our results confirmed that the mechanism of action could be mediated through AMPK and AKT activation, thus enhancing GLUT4 translocation. It was further observed that the compound and crude extract presented substantial inhibitory effects against α-amylase, α-glucosidase, and DPPH. Therefore, in vitro extracts of *W. salutaris* demonstrated good therapeutic efficacy as an antidiabetic and antioxidant agent. To the best of our knowledge, this is the first report describing an iso-mukaadial acetate isolated from *W. salutaris* that is able to activate an insulin signal. Further scientific validation including inhibition studies is essential to understand the therapeutic potential of the plant *W. salutaris* for improving glycaemic control in diabetic subjects and confirm its antidiabetic mode of action. 

## Figures and Tables

**Figure 1 biomolecules-09-00520-f001:**
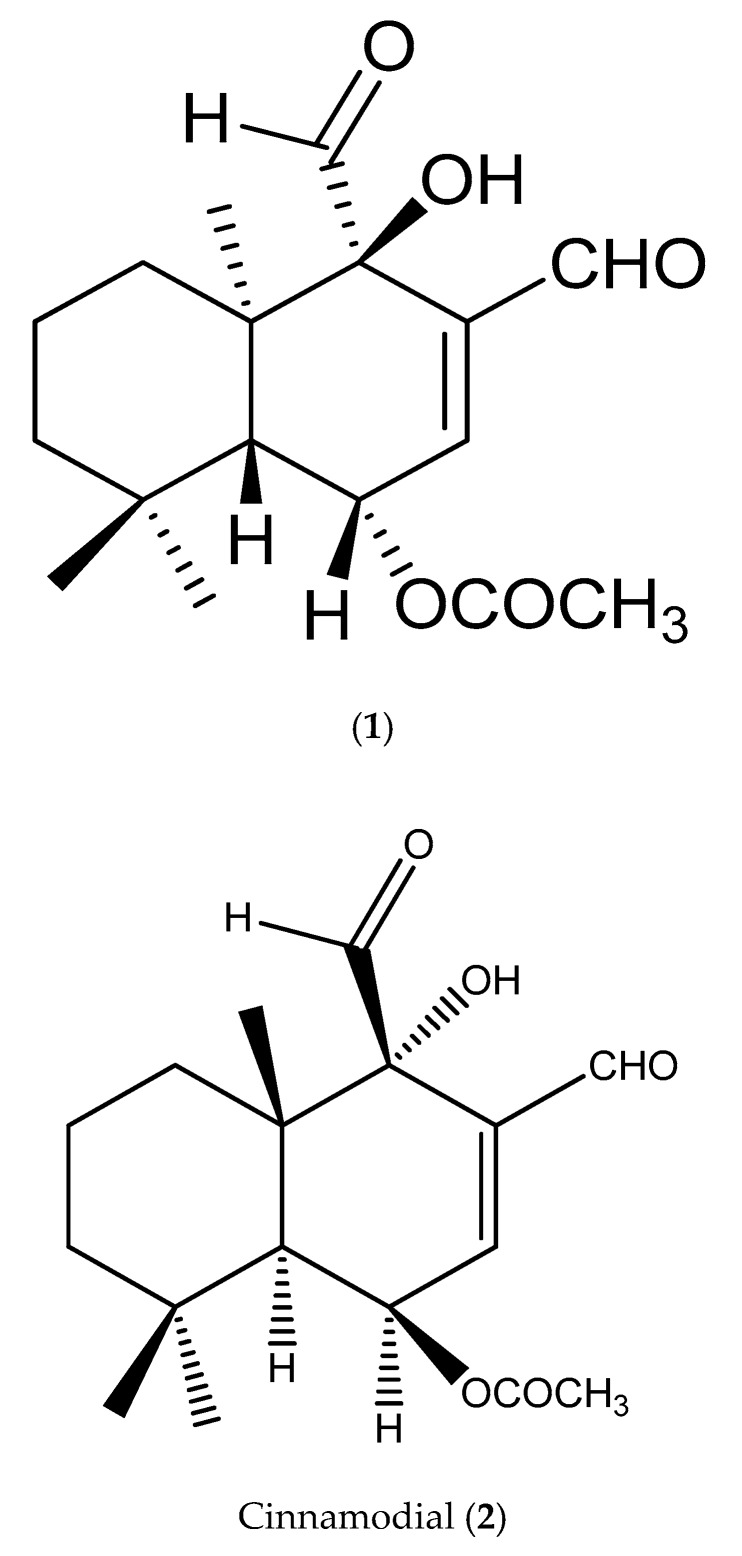
Structure of iso-mukaadial acetate (**1**) (chemical name: (1S,4R,4aR,8aR)-3,4-diformyl-4-hydroxy-4a,8,8-trimethyl-1,4,4a,5,6,7,8,8a-octahydronaphthalen-1-yl acetate) and Cinnamodial (**2**).

**Figure 2 biomolecules-09-00520-f002:**
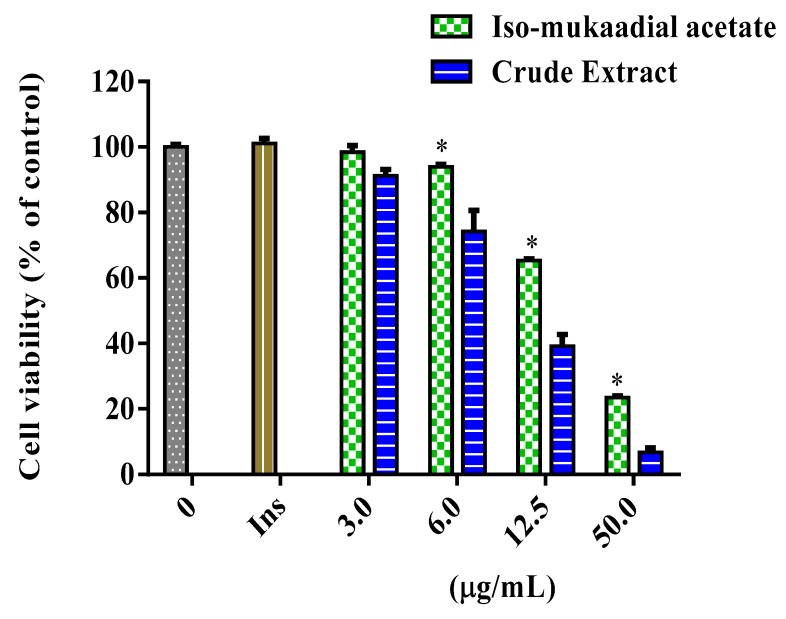
Cytotoxicity of L6 cells after 48-h treatment with iso-mukaadial acetate and crude extract. Cytotoxicity was determined using crystal violet and data expressed as a percentage of the untreated control, * *p* < 0.05.

**Figure 3 biomolecules-09-00520-f003:**
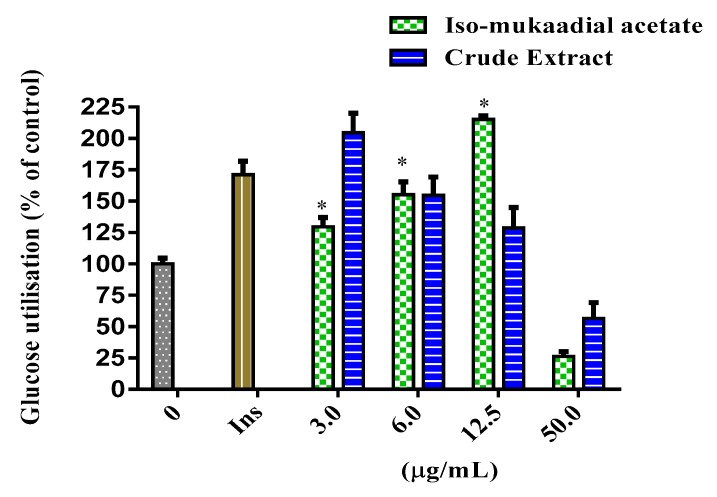
Glucose utilisation in L6 cells after 48-h treatment with iso-mukaadial acetate and crude extract. Glucose utilisation is calculated as the difference in remaining glucose after 2 h incubation between the no cell control and cells with the respective treatments. Data is expressed as a percentage relative to the untreated control, * *p* < 0.05.

**Figure 4 biomolecules-09-00520-f004:**
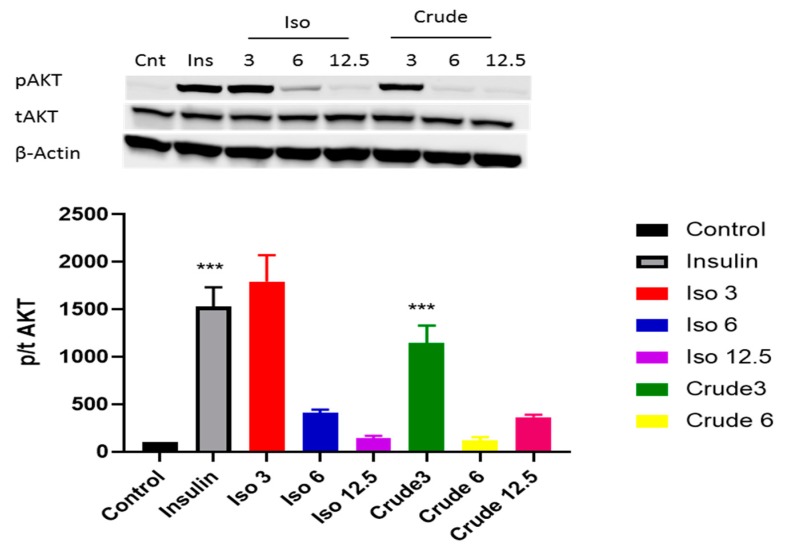
The effect of iso-mukaadial acetate and crude extract on protein kinase B (AKT) protein expression in L6 skeletal muscle cells. Results are expressed as mean ± SEM of three independent experiments., *** *p* < 0.001 versus, Iso 3: iso-mukaadial acetate 3 µg/mL, Iso 6: iso-mukaadial acetate 6 µg/mL, Iso 12: iso-mukaadial acetate 12.5 µg/mL, Crude 3: crude extract 3 µg/mL and Crude 6: crude extract 6 µg/mL.

**Figure 5 biomolecules-09-00520-f005:**
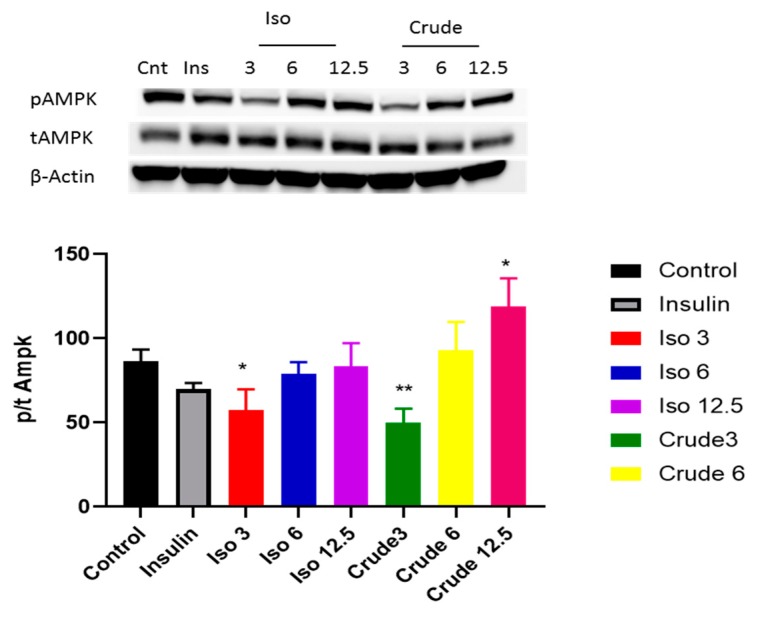
The effect of iso-mukaadial acetate and crude extract on AMP-activated protein kinase (AMPK) protein expression in L6 skeletal muscle cells. Results are expressed as mean ± SEM of three independent experiments. Levels of significance * *p* < 0.05, ** *p* < 0.01 versus, Iso 3: iso-mukaadial acetate 3 µg/mL, Iso 6: iso-mukaadial acetate 6 µg/mL, Iso 12: iso-mukaadial acetate 12.5 µg/mL, Crude 3: crude extract 3 µg/mL and Crude 6: crude extract 6 µg/mL.

**Figure 6 biomolecules-09-00520-f006:**
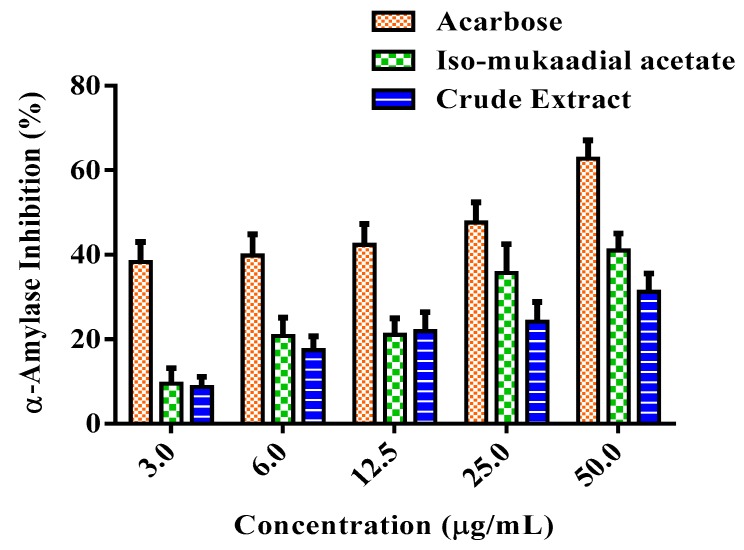
Percentage α-amylase inhibitory effect of a standard compound of acarbose, iso-mukaadial, and crude extract. Values are presented as means, and vertical bars indicate SEM, *p* < 0.05.

**Figure 7 biomolecules-09-00520-f007:**
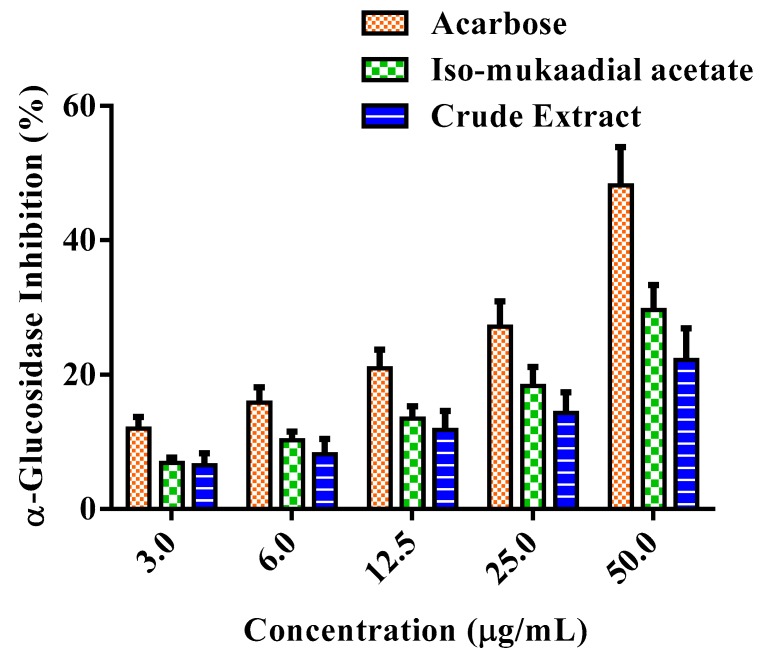
Percentage α-glucosidase inhibitory effect of standard compound of acarbose, iso-mukaadial acetate, and crude extract. Values are presented as means, and vertical bars indicate SEM, *p* < 0.05.

**Figure 8 biomolecules-09-00520-f008:**
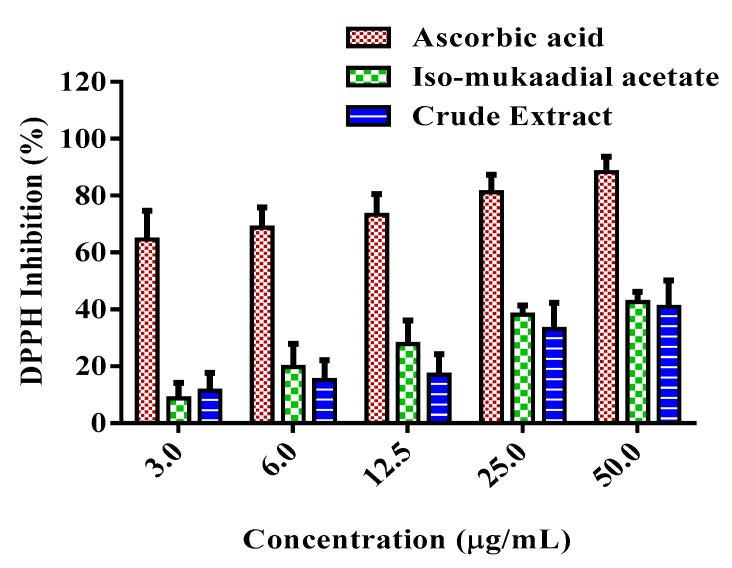
Percentage DPPH inhibition activity of standard compound ascorbic acid, iso-mukaadial acetate, and crude extract. Values are presented as means, and vertical bars indicate SEM, *p* < 0.05.

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
