# Peer review of "Iso-Mukaadial Acetate from Warburgia salutaris Enhances Glucose Uptake in the L6 Rat Myoblast Cell Line"

_biomolecules, 2019, doi:10.3390/biom9100520_

Round 1
Reviewer 1 Report
The author improved the manuscript by considering most of the comments coming from the past revision stage. However, some issues still remain unsolved.
Please improve the aim of your study. This part is totally lacks of any explanations.
Moreover, the letters and numbers in the tables must be checked once again.
I also suggest Moderate English changes required
In my opinion it’s not very clear how characterised chemical the crude extract. How you express their concentrations.
Response: Yes, the IC50 was calculated. Iso-mukaadial acetate 4,6 ug/ml and crude extract 9,36 ug/ml.
Here I still have a question: what standard did you use for quantitative analysis of crude extract?
Author Response
Reviewer 1
Comment 1: Please, improve the aim of your study.
Response: We have added more information on the aim of the study. “Drimane sesquiterpenoids have been reported to have useful biological activities. The main aim of this study was to investigate the antidiabetic potential of the crude DCM extract from the stem bark of Warburgia salutaris (Bertol.f.) Chiov. (Canellaceae) as well as the pure iso-mukaadial acetate which has been investigated for antimalarial activity and to build rich reservoir of pharmacologically established antidiabetic phytoconstituents with specific references to the novel compounds that are affordable which might be of relevance to other low-income and middle-income countries of the world.”
Comment 2: Moreover, the letters and numbers in the table must be checked once again.
Response: We have checked the tables and figures and made some corrections.
Comment 3: I suggest moderate English changes required.
Response: We have made some changes to improve the English of the manuscript.
Comment 4: In my opinion it’s not very clear how characterised chemical the crude extract. How you express their concentrations.
Response: A known weight of the crude extract (solid) was dissolved in the solvent (DMSO) (ml). Similarly, for iso-mukaadial acetate. 1mg of crude solid extract was dissolve in 1 ml of DMSO to get the stock solution and the stock solution was used to prepare serial dilutions.
Reviewer 2 Report
The authors should consider mentioning all responses (where applicable) inside the manuscripts in the appropriate sections.
With due respect, a reviewer asking a question must be because it was not clear or missing! The readers (if accepted) will have similar questions. Thus the authors should consider adding those responses in the manuscript as well, not only for arguing with the reviewer and leave that question unanswered in the manuscript!!
Here I mention a couple of examples of responses, however, the authors should consider explaining or stating this information inside the manuscript.
The authors should state the full question asked by the reviewer.
"So why there is a need to extract this compound?" - Was this full question or statement?
However, the answer is still not clear to me. So there was more study? Which is not included?
Reviewer’s comment: Just for the sake of argument, say, there is a strong basis to chase down this compound. But what is the importance of this compound?
Response: This isomeric compound is important because it has never been isolated before or it was, but was wrongly identified as mukaadial acetate. So, we were anxious to evaluate its pharmacological activities.
Reviewer’s comment: Line # 56 “....pivotal role by removing nearly 90% of glucose” What did
the author mean by “removing”? is it glucose metabolism or oxidation?
Response: glucose metabolism
Author Response
Reviewer 2:
Comment 1: Why there is a need to extract this compound?
Response: The crude extract showed antidiabetic activity. We were interested to know the compounds responsible for the activity. Therefore, the crude extract was fractionated using silica gel column chromatography followed by structural characterization of the isolated compounds. It was then that we discovered the presence of drimane sesquiterpenoids and isolated iso-mukaadial acetate as the main component of the crude extract. Furthermore, drimane sesquiterpenoids have been reported before to have antidiabetic activity.
Comment 2: Line #56 “…pivotal role by removing nearly 90% of glucose” What did the author mean by “removing”? is it glucose metabolism or oxidation?
Response: We have rephrased the sentence by removing nearly 90% of glucose as follows: “enhancing glucose uptake from insulin-sensitive muscle cells through the AMPK/p38 MAPK signaling pathway”. This is a glucose metabolism.
Round 2
Reviewer 1 Report
The manuscript has been improved.
Reviewer 2 Report
The manuscript has been improved. New information is added.
Replies are satisfactory.
This manuscript is a resubmission of an earlier submission. The following is a list of the peer review reports and author responses from that submission.
Round 1
Reviewer 1 Report
The authors have chosen a medicinal plant W. salutaris (Bertol. f.) Chiov.(stembark), which has no history with glucose utilization or improving diabetes. Usually, a basic screening among couple of potential medicinal plants or previous history of using a plant in the diabetes treatment being used in the isolation and further screening. I am not sure in what context/basis the authors have chosen this plant to be tested. The authors have mentioned: “Thus far, there has been no scientific validation for the antidiabetic activity of this medicinal plant.” This cannot be a scientific basis to start a research at the first place. There are billions of plant those are not evaluated for diabetes treatment. The authors should make a strong justification why they have chosen this plant. Also I have other concerns:
Comments:
During fractionation the authors must have produced a number of subtractions. Do the authors investigated or screened those fraction for activity? If yes, then show or state those result. Fraction 2-4 combination produced a white powder, which I am guessing gave the structre of iso-mukaadial acetate. But what happens to other compounds? Why specifically iso-mukaadial acetate was purified to be tested? Is there any HPLC analytical profile for the purification or fraction 2-4 to show that those 3 fractions only had only one compound? Again how the authors ended up by one compound? Did the authors get a better screening activity in this fractions? Just for the sake of argument, say, there is a strong basis to chase down this compound. But what is the importance of this compound? The whole extract showing similar result like isolated compound. In some cases, effects of crude extracts are even better. So why there is a need to extract this compound? Looking at the effects of crude extract, there must be other compounds the extract giving this insulin signalling. I would suggest investigating that or those compounds which will be more potent. Introduction, Line number 52 says: “Medicinal plants as antidiabetic agents are very promising” where is the evidence or any reference? Line # 56 “….pivotal role by removing nearly 90% of glucose” What did the author mean by “removing”? is it glucose metabolism or oxidation? Line # 66: it would be more appropriate to say – “Therefore, screening of some potential natural inhibitors of these enzymes from medicinal plants has received much attention over the years”Reviewer 2 Report
Dear authors please find my sugestions/correction on the manuscript attached.
